# Prospect of Gold Nanoparticles in Pancreatic Cancer

**DOI:** 10.3390/pharmaceutics16060806

**Published:** 2024-06-14

**Authors:** Tianyi Yin, Jingrun Han, Yuying Cui, Dong Shang, Hong Xiang

**Affiliations:** 1Department of General Surgery, The First Affiliated Hospital of Dalian Medical University, Dalian 116011, China; ytydmu@163.com (T.Y.); wslhjr1999@163.com (J.H.); 2Clinical Laboratory of Integrative Medicine, The First Affiliated Hospital of Dalian Medical University, Dalian 116011, China; cuiyuying0220@163.com; 3Institute of Integrative Medicine, Dalian Medical University, Dalian 116044, China

**Keywords:** gold nanoparticles, pancreatic cancer, diagnosis, therapy, safety

## Abstract

Pancreatic cancer (PC) is characterized by its notably poor prognosis and high mortality rate, underscoring the critical need for advancements in its diagnosis and therapy. Gold nanoparticles (AuNPs), with their distinctive physicochemical characteristics, demonstrate significant application potential in cancer therapy. For example, upon exposure to lasers of certain wavelengths, they facilitate localized heating, rendering them extremely effective in photothermal therapy. Additionally, their extensive surface area enables the conjugation of therapeutic agents or targeting molecules, increasing the accuracy of drug delivery systems. Moreover, AuNPs can serve as radiosensitizers, enhancing the efficacy of radiotherapy by boosting the radiation absorption in tumor cells. Here, we systematically reviewed the application and future directions of AuNPs in the diagnosis and treatment of PC. Although AuNPs have advantages in improving diagnostic and therapeutic efficacy, as well as minimizing damage to normal tissues, concerns about their potential toxicity and safety need to be comprehensively evaluated.

## 1. Introduction

Pancreatic cancer (PC) represents a malignancy characterized by poor prognosis and high mortality. In 2020, it was responsible for an estimated 460,000 deaths globally, making it the seventh most deadly malignant tumor [1]. The absence of effective screening approaches for PC and its presentation with mild and nonspecific symptoms, such as abdominal pain, weight loss, jaundice, and digestive problems, leads to the majority of patients being diagnosed in the advanced stages. The delay in discovery complicates treatment and markedly impacts the prognosis [2]. According to 2023 cancer statistics, the 5-year survival rate for PC patients in the United States is less than 12% [3]. Post-operative outcomes indicate that the 5-year survival rate for PC is only 20% [4]. Currently, research on PC is experiencing a critical impasse. On the one hand, PC is challenging to diagnose early due to the lack of early-stage biomarkers and distinct clinical symptoms. Computed tomography (CT) and magnetic resonance imaging (MRI) are frequently used in clinical imaging diagnosis of PC. Research indicates endoscopic ultrasound (EUS) to be more sensitive than CT, particularly for tumors under 3 cm in diameter, and has seen increased application in PC diagnosis in recent years [5]. Nevertheless, the majority of PC cases have metastasized by the time of initial diagnosis, with only 9.7% in a localized stage [2,6]. Therefore, identifying improved diagnostic methods is vital to enhance the prognosis of PC. On the other hand, PC theory depends on the stage of the tumor. Standard treatment for resectable tumors involves adjuvant chemotherapy post-surgery. FOLFIRINOX and nab-paclitaxel-gemcitabine are recommended for patients with metastatic PC [7]. For patients with locally advanced tumors at the borderline of surgical treatment, neoadjuvant chemotherapy or chemoradiotherapy followed by surgical resection is applicable [8]. Targeted therapy and immunotherapy are viewed as promising methods in the ongoing development and trials for PC [4]. In conclusion, the effectiveness of current diagnosis and treatment protocols of PC still require enhancement, necessitating the development of novel diagnostic and therapeutic approaches.

The application of gold nanoparticles (AuNPs) in medicine has attracted significant attention due to their distinctive physicochemical characteristics. AuNPs have made significant strides in fields including photothermal therapy, drug delivery, radiation sensitization, and assisting in the diagnosis of malignant tumors, especially in diagnosing and treating breast and gastrointestinal cancers [9,10,11,12]. Furthermore, AuNPs provide advantages including cost-effectiveness, eco-friendliness, and high biocompatibility. Recently, the exploration of using AuNPs for diagnosing and treating PC has seen a surge in interest. This article aims to deliver a comprehensive review of the advances in AuNPs for the diagnosis and treatment of PC, along with its clinical challenges. In comparison to prior reviews [13,14], this article additionally covers the characteristics, preparation techniques, diagnostic applications in PC, and the safety aspects of AuNPs. Moreover, it amalgamates recent research findings, providing a systematic review of the use of AuNPs in diagnosing, drug delivery, and phototherapy in PC, as depicted in Figure 1.

## 2. Characteristics of AuNPs

AuNPs display a variety of remarkable characteristics, making them highly promising for applications in cancer therapy. Localized Surface Plasmon Resonance (LSPR) constitutes a key feature of AuNPs, characterized by the interaction of conduction electrons with incident radiation, leading to light scattering and absorption [15]. Utilizing the LSPR effect enables AuNPs to efficiently absorb light energy and convert it into thermal energy. This capability is extensively utilized in cancer photothermal therapy for inducing localized hyperthermia and tumor ablation [16,17], while simultaneously enhancing the tumor’s immune response [18]. Adjusting the size of AuNPs enables researchers to optimize their absorption of near infrared (NIR), which is safe and penetrates deep tissues, thereby enhancing therapeutic effectiveness [16].

The enhanced permeability and retention (EPR) effect, proposed by Maeda et al. in 1986, is recognized as a fundamental aspect of nanoparticle-tumor interactions [19]. The EPR effect elucidates why nanoparticles of a certain size tend to accumulate in tumor vessels, attributable to the rapid growth of tumor cells compared to normal tissues, gaps between tumor vascular endothelial cells, and deficiencies in the lymphatic system [20]. Subsequent research into the passive targeting of tumors by nanoparticles has thoroughly incorporated the EPR effect, yielding numerous positive findings. The widely adopted PEGylation process for nanoparticles prolongs their systemic circulation, further enhancing their ability to passively target tumor tissues through the EPR effect [21]. The first commercialized nanodrug, Doxil^®^ (PEGylated liposomal doxorubicin), capitalizes on the EPR effect for passive tumor targeting [22]. Despite its validation in preclinical trials, most clinical treatments that utilize the EPR effect have not met expectations [23]. Many clinical studies have found that nanoparticles only reduce toxicity and have limited improvement in therapeutic efficacy. The EPR effect exhibits significant heterogeneity across different patients, tumor types, tumor sizes, and tumor locations, termed as the heterogeneity of EPR. For example, the EPR effect varies between different solid tumors and even different regions within the same tumor. The EPR effect performs poorly in PC due to the dense extracellular matrix (ECM) and abnormal vascular structures within the tumor microenvironment (TME). This variability in the EPR effect is further exemplified by significant differences between a patient’s primary and metastatic tumors [24]. Typically, the EPR effect is predominantly observed in larger and mature tumors, while it is less effective in newly formed, smaller tumors [25]. Moreover, events such as thrombosis, which decrease blood perfusion, can further modify the EPR effect [26]. In addition to this heterogeneity, design deficiencies in nanoparticles can lead to ineffective EPR outcomes. According to Dr. Maeda, unlike passive targeting, the EPR effect’s tumor targeting is distinct, as evidenced by differing accumulation times within tumor tissues [27]. The half-life of nanoparticles also plays a crucial role. A short half-life means that after conversion to free low molecular weight drugs, they are unable to target and accumulate in tumors for prolonged periods using the EPR effect. Conversely, a long half-life can result in slow drug release, which may consequently lead to poor therapeutic outcomes. Moreover, factors such as the size, shape, surface charge, and surface modifications of nanoparticles significantly influence the efficacy of the EPR effect.

In light of these findings, many studies have devised strategies to augment the EPR effect. These strategies encompass nanoparticle modification and both physical and pharmacological treatments to modify the TME [24,28]. Appropriate design regarding size, shape, half-life, charge, surface characteristics, and biocompatibility of nanoparticles can enhance their EPR effect [29]. Generally, nanoparticles ranging from a few to approximately 100 nanometers in size are seen as more effective at utilizing the EPR effect to target tumors, with effectiveness varying by the tumor type and nanoparticle variety [28]. This size range is crucial for optimizing the EPR effect in targeting tumors, although the ideal size may differ based on the type of tumor and nanoparticle [28]. For the design of nanoparticle half-lives, it is critical that they release drugs at the optimal moment since releasing too early or too late can compromise therapeutic efficacy. Furthermore, designs that leverage hydrophobicity, pH, and hypoxic conditions of the TME ensure nanoparticles release drugs at the proper time [30]. Enhancement of the EPR effect in nanoparticles through physical therapies, such as hyperthermia (HT), photodynamic therapy (PDT), boron neutron capture therapy (BNCT), sonodynamic therapy (SDT), and radiation therapy (RT) is also recognized [27,28,30]. Specifically, PDT leads to the disassembly of endothelial cell microtubules and induces the formation of actin stress fibers, thus increasing gaps within the tumor vascular endothelium and enhancing vascular permeability [30,31]. To combat abnormal tumor blood flow, enhancing the EPR effect can be achieved by increasing vascular permeability, improving blood flow within the tumor, or vascular normalization, such as using vasodilators and vascular active cytokines [27,32]. Moreover, apart from altering nanoparticles and the TME, direct infusion of nanoparticles into tumor arteries allows for enhanced drug targeting and reduces the dosage and side effects of systemic medications, showing promising clinical effectiveness [27].

For nearly forty years, the EPR effect has been considered the mechanism by which nanoparticles penetrate solid tumors. Recently, another mechanism for the accumulation of nanoparticles in tumors has been suggested. Sindhwani and colleagues have concluded from extensive experimental analysis that more nanoparticles likely penetrate tumor tissues through transendothelial routes, with fewer nanoparticles extravasating through inter-endothelial gaps [33,34]. Although this finding presents a different perspective on how nanoparticles enter solid tumors, further experimental validation is needed due to the diverse characteristics of different nanoparticles, which may yield different observations [35]. In conclusion, whether optimizing the EPR effect or exploring new mechanisms, the pathways and mechanisms by which nanoparticles enter solid tumors remain worthy of research and discussion.

## 3. Synthesis of AuNPs

The prevalent fabrication methods for AuNPs are classified into ‘top-down’ and ‘bottom-up’ categories, representing the synthesis from bulk materials and the atomic level, respectively [36]. While top-down synthesis is suitable for mass production, it requires significant investment. In contrast, bottom-up synthesis is distinguished by its low cost, operational simplicity, and excellent scalability [37]. Predominantly, bottom-up synthesis involves techniques such as chemical reduction and biosynthesis. The Turkevich method for synthesizing AuNPs holds a milestone significance in the chemical synthesis of AuNPs. Briefly, the method entails dissolving chloroauric acid (HAuCl_4_) in deionized water, heating it to boiling, and then adding sodium citrate as the reducing agent. By varying temperature, pressure, pH, and sodium citrate concentration, AuNPs with different diameters, parameters, and features can be produced [38]. Later studies have modified the Turkevich method to attain lower variability, enhanced uniformity, and repeatability [39,40,41,42,43,44]. The use of certain reducing agents or stabilizers in the chemical synthesis process, such as sodium borohydride, could potentially be toxic to both individuals and the environment [37,45]. Compared to other methods, the biosynthesis of AuNPs offers environmentally friendly, cost-effective, non-toxic, and highly biocompatible solutions. Several green biosynthesis approaches for synthesizing AuNPs in PC diagnosis and treatment employ plants such as *Borassus flabellifer* L., *Scutellaria barbata*, *Panax notoginseng* leaves, Acai berry, and Elderberry [46,47,48,49]. Extracts from these plants are added to HAuCl_4_ or NaAuCl_4_ solutions and thoroughly mixed. A visible color change occurs during this process, and after adjusting temperature and other parameters, an initial aqueous solution of AuNPs is produced. The AuNPs synthesized via this green method using plants are not only non-toxic and cost-effective but also demonstrate remarkable results in the diagnosis and treatment of PC.

## 4. Applications of AuNPs in the Diagnosis of PC

Accurate staging of PC at the time of diagnosis is crucial for guiding patients toward the most effective treatment strategies [50]. CT angiography, as well as chest and pelvic CT, are utilized for evaluating vascular anatomy and staging the disease. MRI and cholangiography aid in ascertaining if uncertain liver lesions might indicate metastasis and in identifying cancers that CT imaging may not adequately characterize [2]. However, these diagnostic techniques are reliant on the physician’s image-reading skills and experience, potentially resulting in missed diagnoses. Currently, numerous studies are employing nanoparticle-assisted imaging to increase diagnostic sensitivity. Conjugates of AuNPs with F19 monoclonal antibodies significantly aid in the MRI detection of human PC tissues [51]. Darkfield microscopic imaging of PC tissues treated with AuNPs near their maximum resonance scattering (approximately 560 nm) shows distinct positive images in the tumor interstitium, whereas healthy tissues display only sparse isolated nanoparticles. This research offers a promising direction for enhancing the sensitivity of laparoscopic examinations in identifying tumor metastatic sites. In comparison to conventional contrast agents, Gd(III) contrast agents derived from AuNPs enhance the low contrast typically seen in pancreatic imaging. The experiments showed a marked enhancement in pancreatic contrast, enabling clear delineation of the pancreas with a contrast-to-noise ratio over 35:1 [52]. HAuCl_4_ is combined with the MRI contrast agent dotarem and then formed into a contrast agent–AuNPs conjugate using a lactose-modified chitosan polymer. In vivo experiments demonstrate that the conjugate possesses an effective T1 high signal and features a reduced clearance time [53]. 5B1 is a fully humanized monoclonal antibody that targets the CA19-9 antigen, commonly overexpressed in PDAC [54]. Researchers utilized AuNPs integrated with the 5B1 antibody, clodronate liposomes, and 89Zr for innovative PET/CT imaging in in vivo PDAC models. AuNPs labeled with 5B1 demonstrated an accumulation in subcutaneous and orthotopic PDAC that was 4–7 times greater than that in the IgG control group [55]. AuNPs notably increase the sensitivity of radiographic diagnosis, surpassing the constraints of conventional CT and MRI, thus providing a promising approach for more precise diagnosis and staging of PC.

Fluid-based research biomarkers, such as free DNA, exosomes, and circulating tumor cells, are also applicable in tumor auxiliary diagnosis, treatment response monitoring, and assessing resistance to treatments [56,57,58,59]. Many researchers utilize PC-specific antibodies in conjunction with AuNPs to create antibody–AuNPs conjugates (Ab-AuNPs), thereby increasing detection sensitivity. Microfluidic technology, frequently used in detecting circulating tumor cells, is known for its high sensitivity and specificity. A novel lateral filter array, equipped with AuNPs carrying anti-EpCAM antibodies, is capable of capturing circulating tumor cells. In both in vitro PC cell line and clinical sample experiments, this array notably enhances the capture efficiency of CTCs [60]. The new tyrosine kinase PEAK1 is found to be overexpressed in PDAC and pancreatic intraepithelial neoplasia [61]. A paper-based immunosensor exploits the catalytic properties of AuNPs in dye degradation to colorimetrically detect the PC biomarker PEAK1 [62]. The sensitivity of this detection approach is ten times higher than that of non-signal amplified AuNPs immunochromatography. AuNPs, when conjugated with anti-CA19-9 antibodies, are capable of detecting the PC biomarker CA19-9 in plasma efficiently [63]. This technique is not just highly sensitive, but it is also anticipated to quantitatively assess CA19-9 levels for future treatment monitoring. Lin et al. developed an amplified time-resolved lock nucleic acid sensor with AuNPs for the selective electrochemical detection of K-ras mutations in PC. The sensor shows high specificity and sensitivity, distinguishing between wild-type and mutation-type K-ras DNA, with an estimated detection limit of 0.5 fM, providing a novel diagnostic perspective for K-ras point mutations in PC [64]. Research indicates that the lncRNA HOXA distal transcript antisense RNA (HOTTIP) is aberrantly elevated in PC [65], making it an effective circulating biomarker for PDAC diagnosis. Lou et al. devised a colorimetric technique that combines reverse transcription coupled with loop-mediated isothermal amplification and the aggregation of positively charged AuNPs for detecting HOTTIP [66]. Leveraging the characteristics of AuNPs, the researchers developed a sensitive, stable, and portable platform for mRNA detection. Using catalytic hairpin assembly and an Au enhancer buffer (HAuCl_4_/NH_2_OH·HCl) to boost gold deposition, they doubled the amplification of the PC mRNA GPC1 signal, effectively identifying the PC cell line AsPC-1 [67]. AuNPs, when conjugated with specific antibodies, substantially enhance the detection efficiency of circulating tumor cells, PEAK1, CA19-9, and HOTTIP, paving the way for new opportunities in early detection and treatment monitoring of PC.

## 5. Applications of AuNPs in the Treatment of PC

### 5.1. Drug Delivery

Systemic chemotherapy regimens such as FOLFIRINOX (5-fluorouracil, folinic acid, irinotecan, and oxaliplatin) and gemcitabine plus nab-paclitaxel continue to be the main treatments for patients with advanced PC [4]. Numerous studies have confirmed the clinical effectiveness of chemotherapy in treating PC [68]. The dense connective tissue proliferation and immunosuppressive traits within the TME in PC contribute to the less-than-optimal outcomes of chemotherapy [69,70,71]. Furthermore, standard chemotherapy regimens for PC are known to have substantial side effects. In an effort to bypass the drawbacks of traditional chemotherapy, the synthesis of chemotherapy drugs with nanoparticles into polymers for targeted delivery to PC is gaining traction as a novel area of interest. 

The pathways for drug delivery to PC by AuNPs can be divided into passive and active targeting. The EPR effect is the key mechanism behind AuNPs’ passive targeting. Many experiments have successfully improved the EPR effect of AuNPs by modifying their diameter, shape, and surface chemical properties, thus achieving substantial passive targeting outcomes [72]. A common method involves the PEGylation of AuNPs to prolong their systemic circulation time [21]. AuNPs with smaller diameters are found to exhibit greater accumulation in tumors [72]. However, the dense extracellular matrix (ECM) and complex TME of PC can significantly reduce the EPR effect [73,74]. In response to this challenge, many researchers use phototherapy to modify the TME of PC, facilitating easier passage of AuNPs through the ECM and enhancing drug accumulation.

Furthermore, distinct from passive targeting, AuNPs can actively target tumor cells through conjugation with antibodies, proteins, peptides, nucleic acid aptamers, carbohydrates, and small molecules, and be selectively uptaken by tumors via receptor-mediated endocytosis [75,76]. Chitta and colleagues pioneered the use of cetuximab to actively target GEM-loaded AuNPs to PC, marking the first study of antibody-mediated active targeting of AuNPs [76,77]. Zoë et al.’s review thoroughly summarizes studies related to the active targeting of AuNPs [76]. In active targeting therapy for PC, nanoparticles are directed towards targets such as EGFR, urokinase plasminogen activator receptor (uPAR), transferrin, ERBB2, CA125, and stem cell markers like epithelial cell adhesion molecule (EpCAM), CD44, and CD133 [78]. After reaching the tumor tissue through either passive or active targeting, drug-loaded AuNPs release their drugs via pH alterations, enzyme-triggered reactions, or by utilizing the LSPR effect in photothermal and ultrasound applications [79]. Figure 2 shows the mechanism of action of AuNPs in the drug delivery for PC.

GEM serves as a primary chemotherapy agent in treating advanced PC and is deemed the gold standard for single-agent therapy in this cancer [80]. However, the therapeutic efficacy of GEM in the treatment of PC falls short of expectations [81]. To improve its therapeutic efficacy, numerous nanodelivery systems such as liposomes, polymeric nanoparticles (albumin and chitosan), etc., have been explored for GEM-based treatment of PC [82]. Lizhou et al. developed a scheme for ultrasound-targeted microbubble destruction (UTMD)-assisted targeted delivery of GEM using AuNPs for treating PC. UTMD enhances the permeability of cancer cells, facilitating the uptake of drugs [83]. During in vitro experiments, under UTMD assistance, AuNPs release GEM slowly, yet cytotoxicity increases over time, leading to a higher rate of cell apoptosis. In vivo experiments revealed that the conjugate group also attained more notable tumor suppression outcomes [84]. A drug delivery system that includes polyethylene glycol (PEG), cetuximab, and AuNPs carrying GEM yielded favorable outcomes in in vitro experiments. With a 10 μM concentration of AuNPs conjugates, the cell survival rate for PC cells Panc-1 and AsPC-1, and stellate cells CAF-19, was 30%, showing lesser toxicity to healthy human pancreatic cells [85]. The targeted delivery of GEM via AuNPs, along with glutathione, notably reduces the viability of PC cells. After treatment of Panc-1 cells with the conjugate, their viability dropped to approximately 25% [86]. By combining GEM with AuNPs, researchers have enhanced the drug’s cellular uptake and the apoptosis rate of tumor cells in the nanodelivery system. The experiments demonstrate that AuNPs hold substantial potential in boosting the chemotherapeutic impact of GEM on PC.

The use of AuNPs in conjunction with targeted drugs also yields affirmative outcomes in PC treatment. Afatinib irreversibly binds to the intracellular tyrosine kinase domains of the ErbB receptor family [87]. Research has shown that the combination of afatinib and GEM possesses significant potential in the treatment of PC [88]. In the PC cell line S2-013, combining PEGAuNPs with afatinib was five times more efficacious in suppression than afatinib alone (with half maximal inhibitory concentration [IC50] values being 0.103 ± 0.001 vs. 0.50 ± 0.02, respectively) [89]. Varlitinib, a reversible small-molecule pan-HER inhibitor, targets EGFR, HER2, and HER4 [90]. Experiments involving targeted drug delivery to the PC cell line MIA PaCa-2 using AuNPs conjugated with varlitinib yielded significant outcomes. The IC50 was 2.5 times lower with AuNPs conjugates compared to using varlitinib alone. With equivalent concentrations of varlitinib, AuNPs conjugates demonstrated increased cytotoxicity towards MIA PaCa-2 cells [91]. In vitro, the release of doxorubicin and varlitinib linked with PEGAuNPs was more prolonged in 48 h than free drugs, augmenting the inhibition of PC cell lines S2-013 and MIA PaCa-2 by 2–4 times. The conjugates also diminished the drug’s toxicity towards bystander cells hTERT-HPNE [92].

AuNPs are also capable of delivering various drugs for PC treatment. Bortezomib (BTZ), a boronic acid-based proteasome inhibitor, is typically used to treat multiple myeloma [93]. Research indicates that BTZ causes apoptosis in PC cells, potentially linked to ceramide production in primary and transformed PC cells [94]. The combination of BTZ and PEGAuNPs in treating PC cells leads to increased mass transfer across cell membranes, facilitated by augmented cellular uptake and endosome formation, thereby enhancing the cytotoxic effect of BTZ at extremely low concentrations (0.1–1.0 nM) [95]. The free BTZ requires a 63-fold higher concentration than PEGAuNPs-BTZ conjugate to attain comparable cytotoxicity [96]. Epigallocatechin-3-Gallate (EGCG), a major polyphenolic component of green tea, suppresses PC cell growth, invasion, and migration by inhibiting the Akt pathway and the epithelial-mesenchymal transition [97]. Conjugates of AuNPs with EGCG not only inhibit the growth of BxPC3 cells, but also preserve the antioxidant properties of EGCG [98].

In addition to drug delivery, AuNPs also increase drug sensitivity in PC cells through mechanisms such as inhibiting epithelial–mesenchymal transition, stemness, and mitogen-activated protein kinase signaling, and reducing tumor fibroblast proliferation, thus boosting chemotherapy effectiveness [99,100]. Targeting the dense stroma surrounding PC, nanoparticles equipped with collagenase are capable of degrading the collagen components of the PC matrix, thus enhancing the efficacy of tumor-targeted therapies [101]. In vitro studies show that AuNPs reduce the tumorigenic potential of Panc-2 and MIA PaCa-2 cells. In combination therapy with GEM, AuNPs suppress epithelial–mesenchymal transition, stemness, and mitogen-activated protein kinase signaling in PC cells, resulting in a marked decrease in cell colony formation [100].

### 5.2. Phototherapy

Phototherapy comprises both photothermal therapy (PTT) and PDT. PDT relies on the interaction between photosensitizers, light, and oxygen to generate cytotoxic reactive oxygen species (ROS), which lead to the death of cancer cells. Conversely, PTT employs NIR to elevate tissue temperature, thus directly annihilating cancer cells via thermal effects [102]. Within the realm of nanomedicine, phototherapy presents a vast potential, as it has shown notable antitumor activity when combined with chemotherapy, immunotherapy, and radiotherapy. The characteristic of LSPR is an optical phenomenon, specifically, the interaction between surface electrons in the conduction band and incident light [103]. AuNPs exploit their LSPR effect to absorb specific wavelengths of light and convert this energy into heat. This process selectively increases the temperature of certain tissues, leading to protein denaturation and swift cell death [102,104]. Irreversible cell damage occurs when tissues are subjected to thermotherapy temperatures (above 42 °C) [105]. Table 1 presents the research parameters and results of AuNPs in phototherapy.

In the initial research on the phototherapy of AuNPs, Guo et al. treated Panc-1 cells with nanoparticles that had an iron oxide core and a gold shell, subsequently exposing them to laser irradiation at 7.9 W/cm^2^. The application of cellular MRI techniques revealed a notable decrease in tumor cell proliferation, which varied in a dose-dependent manner with nanoparticle concentration [106]. Kim and colleagues then developed branched AuNPs, synthesized from deoxycholic bile acids, enabling these nanoparticles to absorb higher energy NIR for effective photothermal treatment [107]. Subsequent in vitro experiments employing NIR irradiation on BxPC3 human PC cells resulted in temperatures swiftly rising to 50 °C, achieving a cell mortality rate as high as 90% within three minutes. Further in vivo research showed that photothermal therapy using branched AuNPs was able to elevate the temperature of tumor tissues to 60 °C in 6 min, leading to the dissolution of nuclei in PC cells without evidence of tumor recurrence. Subsequently, Hui and his team developed AuNPs carrying the U11 peptide for actively targeting pancreatic tumors and the PDT agent CRQAGFSL-5-ALA, facilitating combined PTT/PDT treatment of PC under confocal laser endomicroscopy [108]. This active targeting strategy enhanced the concentration of AuNPs in PC, minimizing harm to healthy tissues. Moreover, the combination of PTT/PDT treatments was found to demonstrate significant synergistic effects, with the treated mice exhibiting higher survival rates, lower cell viability, and increased reactive oxygen species (ROS) production compared to controls. Additionally, the NFL-TBS.40-63 peptide (BIOT-NFL) has been shown to be capable of destroying the microtubule network in targeted glioma cancer cells. By leveraging the properties of BIOT-NFL, Spadavecchia’s group utilized AuNPs equipped with BIOT-NFL for the treatment of PC. In this context, MIA PaCa-2 cells treated with BIOT-NFL-PEG-AuNPs demonstrated a higher internal concentration of AuNPs and a more significant decrease in cell vitality post-phototherapy than those treated with PEG-AuNPs [109]. Furthermore, BIOT-NFL-PEG-AuNPs significantly raised the levels of serum IL-6, IFN-γ, and TNF-α, thereby bolstering the immune system’s capacity to suppress PC [109].

The limited depth penetration of NIR in PTT has led to the emergence of interventional photothermal therapy (IPTT) as a novel strategy for the treatment of deep-seated tumors. Hu et al. developed AuNPs that specifically target PC with anti-urokinase plasminogen activator receptor (uPAR) antibodies, thus employing IPTT to treat deeper layers of PC. IPTT offers a more precise eradication of deep-seated PC compared to Iodine-125 (125I) interstitial brachytherapy, resulting in reduced damage to healthy tissues and lower overall toxicity [110]. Honeycomb-like AuNPs (HGNs)-mediated interventional photothermal-near-field radiation therapy (IPT-BT) demonstrates superior synergistic antitumor properties. The in vitro studies on SW1990 and Panc-1 cell lines have shown that HGNs-treated cells exhibited fewer active cell colonies post X-ray exposure compared to untreated ones; cells in the HGNs + PT-RT group exhibited significantly higher late apoptosis rates than controls [111]. Furthermore, in vivo research has indicated that synergistic treatment with HGNs-based IPT-BT aids in eradicating deep-seated tumors and alleviating hypoxia-associated BT resistance, with hemoglobin levels rising in the HGNs + IPTT group upon laser exposure.

The wavelength of NIR plays a critical role in determining penetration and therapeutic efficacy. The NIR wavelengths most frequently studied and applied are NIR-I (750–900 nm) and NIR-II (1000–1700 nm). NIR wavelengths at the longer end of the spectrum possess deeper tissue penetration capabilities, higher radiation thresholds, and increased tissue tolerance [112]. One study compared the impact of two distinct wavelengths on the photothermal treatment of pancreatic tissues. This research revealed that, under identical conditions, the temperature generated by AuNPs at 808 nm was 200% higher than at 1064 nm, resulting in less damage to adjacent normal tissues [113]. Zhang et al. utilized perfluorocarbon (PFC) as an oxygen carrier, aiming to replenish oxygen in the hypoxic environment of PC for PDT [114]. Gold nanorods carrying PFC and DOX were directed towards PC, initially irradiated with the deeper penetrating NIR-II (980 nm) to emit oxygen, leading to engorgement, followed by the release of silicon phthalocyanine (SiPc) with an extinction peak at 680 nm and DOX into PC tissues, and culminating in a PDT treatment using 680 nm NIR. This sequential application of the two NIR types nearly entirely eradicated the mouse tumors, contrasting with less effective outcomes when the sequence was reversed or when only one type of NIR was used. This demonstrates the crucial importance of the NIR wavelength, with the stronger penetration of NIR-II suggesting a new research direction.

**Table 1 pharmaceutics-16-00806-t001:** Studies and results of AuNPs in phototherapy for PC.

Nanoparticles	Radiate Time (Min)	Laser Power Density (W/cm^2^)	The Wavelength of Laser (nm)	Outcome	Cell Lines	Ref
Iron-oxide core/gold-shell nanoparticles	5	7.9	808	Photothermal ablation of Panc-1 cells demonstrated an effective treatment response	Panc-1	[99]
cRGD-branched GNPs	5	1.4	808	Tumors were effectively ablated, without any observation of tumor recurrence	BxPC3	[100]
AuS-U11	5	2	750	Provided better synergistic therapeutic effects against pancreatic tumors	Panc-1	[101]
BIOT-NFL-PEG-AuNPs	15	0.5	808	The vitality of tumor cells significantly decreased	MIA PaCa-2	[102]
gold nanoshells	6	2	808	IPTT offers a more precise eradication of deep-seated PC compared to 125I interstitial brachytherapy	SW1990	[104]
honeycomb-like GNPs	5	2	808	Helpful for eliminating the deep tumors and improving hypoxia-associated BT resistance	Panc-1	[105]
gold nanorods	1	2–5	808/1064	Under 808 nm laser irradiation, tissue heats up slowly, demonstrating selective tissue heating capability	_	[107]
PSPP-Au980-D	5/5	0.1/0.05	980/680	The sequential application of the two NIR types nearly entirely eradicated the mouse tu-mors	MIA PaCa-2	[108]
GEM–polymer conjugate NPs	1	1.4	640	The polymer-bound GEM and the GNPs exhibit a synergistic effect	MIA PaCa-2	[109]
GNPs-pD-PTX-PLGA-MS	3	2	808	Enhanced apoptosis and downregulation of antioxidant enzymes	Panc-1	[111]
gold nanoshells	3	4	808	Demonstrated the synergistic effect of photothermal therapy and chemotherapy	MIA PaCa-2/Panc-1	[112]
Tf-GNRs	3	0.5	808	Laser irradiation obviously induced the blood perfusion and extravasation in tumor areas	MIA PaCa-2	[113]
gold nanocages	5	1	808	NO improves the effectiveness of GEM chemotherapy through vasodilation in tumor tissues	SW1990	[114]

Combining phototherapy with chemotherapy significantly enhances cytotoxic effects against PC cells [115]. A contributing factor to the dense extracellular matrix of PC, which impedes chemotherapeutic drug delivery to tumor tissues, is identified as a key factor in the poor response to chemotherapy. Utilizing NIR, AuNPs are able to accurately release drugs and modify the cancer cell membrane’s permeability, thus enhancing chemotherapeutic drug absorption, improving treatment efficiency, reducing dosage, and lessening chemotherapy’s side effects [17,116]. This combined approach of AuNPs-based phototherapy with chemotherapy for PC demonstrates their synergistic impact. Specifically, for MiaPaCa-2 PC cell lines, the IC50 was almost two times lower when treated with GEM-loaded AuNPs following NIR irradiation, compared to direct drug delivery via AuNPs [115]. Moreover, PTX-carrying AuNPs, post-NIR irradiation, exhibited triple the cytotoxicity against a control group without NIR, along with increased ROS generation and reduced expression of antioxidant enzymes [117]. Innovatively, Poudel et al. created gold nanoshells combining BTZ and GEM chemotherapy with photothermal therapy, using low-power NIR for accurate drug release and high-power lasers for direct tumor cell destruction via photothermal effects. In comparison, compared to control groups treated with either photothermal therapy alone or drug delivery alone, the combination therapy led to a significantly higher rate of cell apoptosis [118]. Exploiting PTT, Zhao et al. leveraged its capacity to boost blood flow and microvascular permeability in tumor cells, thereby enhancing the chemotherapeutic effectiveness of GEM when combined. The Transferrin-coated rod-like mesoporous silica gold nanoshell NPs (Tf-GNRS) actively target PC. Following NIR exposure, increased tumor blood perfusion significantly enhances chemotherapeutic drug accumulation in PC, effectively suppressing the tumor [119]. Furthermore, Zhang et al. utilized nitrogen oxide (NO) for its ability to induce tumor vasodilation and normalize tumor vessels, in synergy with PTT, to boost the efficacy of GEM treatment for PC. The Au nanocages carrying L-arginine (L-Arg) generate NO due to increased ROS levels within the TME. After NIR irradiation, there’s a notable increase in tumor permeability and deep-layer drug accumulation, leading to significant tumor suppression [120].

To conclude, AuNPs have extensive applications in the PTT treatment of PC. By utilizing passive/active targeting by AuNPs, the precise heating of tumor tissues effectively leads to the destruction of tumor cells. Specifically designed for deep-seated tumors beyond the reach of NIR, IPTT has proven to yield favorable therapeutic results. By leveraging PTT’s capacity to enhance tumor blood perfusion and improve the TME, along with AuNPs that are loaded with chemotherapeutic drugs aiming at targeting tumor cells to increase drug concentration in tumor tissues, the combination of PTT and chemotherapy has been shown to achieve notable effectiveness. Photothermal-immunotherapy is gaining increasing attention recently. After undergoing PTT, thermal injury to tumors significantly alters the TME and releases tumor antigens, thereby boosting tumor immunogenicity. The synergy between this approach and immunotherapy yields optimal treatment outcomes [121,122]. Looking ahead, the future of PTT research is expected to focus on the integration of active targeting, chemotherapy, and immunotherapy.

### 5.3. Radiofrequency Therapy

The use of radiofrequency ablation for treating inoperable PC is on the rise [123]. Nonetheless, the non-selective and invasive characteristics of current radiofrequency therapy may lead to patient discomfort. Nanoparticles can serve as a substitute for radiofrequency probes, selectively targeting tumor sites and reducing patient discomfort. Radio waves, unlike NIR-mediated PTT, can travel through objects with minimal absorption, hence they have enhanced biosafety [124]. The combination of AuNPs with radiofrequency fields in cancer treatment creates intense heat within the cells, leading to necrosis or cell death, with little to no harm to surrounding cells or tissues [125].

Treating Panc-1 cells with cetuximab-conjugated AuNPs and subjecting them to a 200 W, 13.56 MHz radiofrequency field for 5 min resulted in Panc-1 cell viability dropping to 39.4 ± 8.3%, with no harm to neighboring Cama-1 cells [126]. Christopher’s team applied a 13.56 MHz external radiofrequency field on Hep3B and Panc-1 cell lines treated with AuNPs at a concentration of 67 μM/L. The death rate in these cells was significantly higher at all points compared to the control, unlike cells that only received the same frequency of external radiofrequency irradiation, which showed no notable cytotoxicity [127]. In another research, in vivo tests were performed to ascertain the anti-PC efficacy of AuNPs. Researchers treated mice implanted subcutaneously with Panc-1 and Capan-1 using AuNPs conjugated with cetuximab and PAM4 antibodies. Post-radiofrequency irradiation, the xenografted pancreatic tumors were notably damaged. Even though AuNPs concentrations rose in the mice’s liver and spleen, no apparent signs of treatment toxicity were observed throughout the study [128]. Table 2 shows the study parameters and outcomes of AuNPs in radiofrequency therapy for PC.

### 5.4. Radiotherapy

In cases of inoperable PC, chemotherapy is often used in conjunction with traditional fractionated external beam radiotherapy [129]. Traditional radiation therapy tends to heavily damage normal tissues around the tumor. Radiation therapy guided by AuNPs as radiosensitizers focuses the treatment on tumor tissues and enhances the efficacy of radiation therapy. During radiation therapy, AuNPs exhibit characteristics like producing ROS and locally heating the tumor tissues [130]. A study demonstrated that AuNP-molecularly imprinted polymer microgels (Au-MIP microgels), used as radiosensitizers for PC, significantly inhibited tumor growth in mice injected with these microgels compared to control mice injected with phosphate-buffered saline during X-ray irradiation [131]. Abdulaziz et al. employed AuNPs to enhance radiation therapy in a 3D in vitro tumor model comprising tumor-associated fibroblasts and MIA PaCa-3 PC cells. The combined use of AuNPs and radiation therapy resulted in a significant reduction in tumor size and cell proliferation, with increased DNA double-strand breaks in both co-culture and single-culture groups, showing AuNPs’ effective radiosensitizing capability [132].

Using docetaxel (DTX) and a lipid nanoparticle-encapsulated DTX prodrug (LNPDTX-P), the authors found that the treated tumor samples exhibited twice the AuNP uptake as control samples in both in vivo and in vitro settings [133]. The combination of ultrasmall AuNPs (USNPs) with a cisplatin precursor enhances the efficacy of radiation therapy. When exposed to ionizing radiation, the combined application of USNPs and a cisplatin precursor delays the DNA damage response induced by ionizing radiation, leading to apoptosis in PC cells [134]. There is growing interest in targeted alpha particle radiation therapy for cancer, with research demonstrating its significant impact on both the diagnosis and treatment of PC [135,136]. An experiment using AuNPs for targeted delivery of 211At in adjunct radiation therapy showed prolonged retention of 211At in PC tissues, indicating substantial anti-PC activity [137]. Table 3 shows the study outcomes of AuNPs in radiotherapy for PC.

## 6. Safety of AuNPs in the Treatment of PC

While AuNPs show substantial potential in medicine, their potential toxicity and safety concerns deserve careful consideration. The article previously referenced experiments assessing AuNPs toxicity, including the addition of bystander cells in vitro and monitoring AuNPs accumulation or reactions in other organs in vivo. Across these studies, no marked toxic effects of AuNPs were detected. Other research has identified potential safety concerns with AuNPs in normal tissues or cells in both in vitro and in vivo settings. Lopez-Chaves’ experiments revealed that AuNPs damage DNA, lipids, and proteins, with smaller-sized AuNPs causing more severe damage [138]. For example, 13 nm diameter PEG-AuNPs have been shown to induce acute inflammation and apoptosis in mouse livers. Post-injection, AuNPs remain for an extended period in the liver, spleen, and bloodstream [139]. In contrast to those measuring 20 nm and 50 nm, 5 nm AuNPs inflict dose-dependent DNA damage and generate ROS. In vivo, 5 nm AuNPs demonstrated considerable embryotoxic damage [140]. The female ovulation cycle must be considered when utilizing nanoparticles. The application of nanoparticles during mice ovulation results in nanoparticle accumulation in the ovaries and uterus being double that of non-ovulatory periods [141]. The excessive buildup of nanoparticles in the ovaries and uterus could potentially impact the reproductive system. Nanoparticles could selectively stimulate tumor cell growth. Nanoparticles with a small diameter are capable of activating the protein kinase B (AKT) and extracellular signal-regulated kinase (ERK) pathways, enhancing cell growth through coupling with EGFR [142]. The research indicates that despite AuNPs’ optimistic application prospects, a comprehensive evaluation of their toxicity and safety is crucial prior to further clinical use. As Khlebtsov and colleagues noted in their paper, AuNPs may present potential risks to humans, yet this does not imply all AuNPs are hazardous, and each new variety should undergo stringent safety testing [143].

## 7. Conclusions and Perspectives

This review emphasizes the diverse applications of gold nanoparticles (AuNPs) in overcoming the challenges of diagnosing and treating pancreatic cancer (PC). PC remains a significant obstacle in oncology, attributed to its delayed diagnosis and limited treatment outcomes. Nanotechnology has shown great promise in enhancing the diagnosis of PC, delivering chemotherapy drugs, and utilizing phototherapy, among other applications. The increasing focus on AuNPs in the treatment of PC is attributed to their advantages such as high biocompatibility, the potential for green synthesis, stability, and low toxicity. By utilizing passive or active targeting methods combined with specific receptors, AuNPs enable the precise delivery of chemotherapy drugs while also mitigating their side effects. Moreover, the combination of drug delivery and phototherapy can significantly improve blood flow and drug permeability in PC, thus boosting the efficacy of chemotherapy. The LSPR characteristics of AuNPs play a critical role in their application in phototherapy for PC. Recent studies have explored the issue of phototherapy’s limited impact on deep-seated tumors through interventional techniques or by adjusting NIR wavelengths. Furthermore, AuNPs have a marked impact on radiosensitization and radiotherapy in PC, reducing the discomfort, harm to adjacent healthy tissues, and systemic adverse effects associated with invasive therapies.

Despite this, the potential toxicity and safety issues related to AuNPs warrant further investigation. The long-term consequences of AuNPs excessively accumulating in organs like the liver and kidneys are still not fully understood. Across different studies, the size and surface modifications of AuNPs vary, which may lead to side effects of differing severities. Addressing this issue requires comprehensive preclinical and clinical studies to establish the safety profiles of various kinds of AuNPs. In using AuNPs for the treatment of PC with diverse modifications and structures, it is essential to conduct a thorough examination of side effects and to perform a careful assessment of the overall benefits relative to these side effects.

Integrating different AuNPs therapeutic methods could represent a promising future research pathway. For example, synergies have been observed in the conjoint use of AuNPs for chemotherapy drug delivery and phototherapy. Phototherapy has been shown to modify the dense ECM of PC, thereby improving its blood flow. AuNPs not only enable precise delivery of chemotherapy drugs but also enhance the drug’s permeation into the tumor. Additionally, to address the challenge of PC’s depth beneath the skin, the exploration of NIR-II, known for its superior tissue penetration, is steadily growing. Presently, research into utilizing AuNPs for supplementary immunotherapy in PC is still emerging. However, PTT/PDT not only modifies the TME of PC but also increases tumor immunogenicity and enhances immune cell infiltration. Combining this approach with immunotherapy could lead to significant therapeutic outcomes. In conclusion, despite the challenges ahead, AuNPs have significant potential to revolutionize the diagnosis and treatment of PC. Building on the existing foundation and addressing future challenges with innovative approaches, the prospects for diagnosing and treating PC appear promising, offering hope for improved patient outcomes and quality of life.

## Figures and Tables

**Figure 1 pharmaceutics-16-00806-f001:**
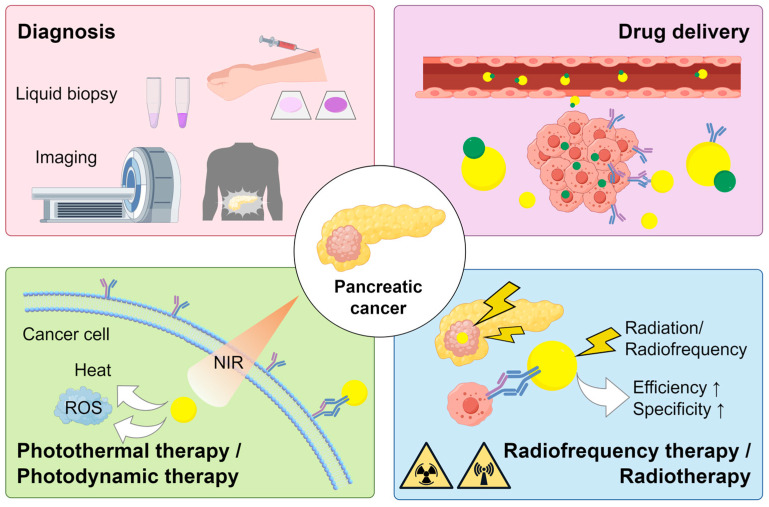
Schematic illustration of the application of AuNPs in the diagnosis and treatment of PC. NIR: near infrared; ROS: reactive oxygen species; “↑” signifies that AuNPs can improve the specificity and efficiency of radiofrequency therapy and radiotherapy in PC. By figdraw.

**Figure 2 pharmaceutics-16-00806-f002:**
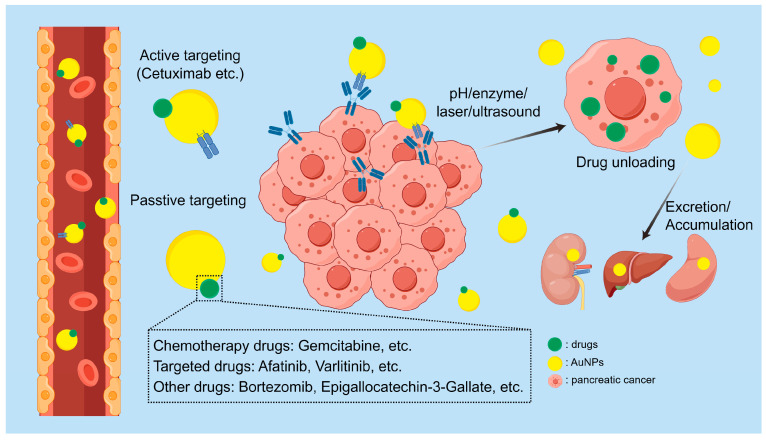
Schematic illustration of the mechanism of drug delivery by AuNPs in PC. AuNPs target tumor tissues through active and passive targeting mechanisms. Drugs are released from AuNPs via alterations in pH, enzymatic reactions, laser irradiation, or ultrasound. Post-drug release, AuNPs are excreted or may accumulate in organs such as the kidneys, liver, and spleen. By figdraw.

**Table 2 pharmaceutics-16-00806-t002:** Studies and results of AuNPs in radiofrequency therapy for PC.

Nanoparticles	Operating Frequency (MHz)	Operating Power (W)	Particle Diameter (nm)	Outcome	Cell Lines	Ref
AuNP-C225-AF647	13.56	200	20	Radiofrequency fields show selective cytotoxic effects on Panc-1 without damaging bystander Cama-1	Panc-1/Cama-1	[126]
AuNP	13.56	200–1000	5	AuNP inflict fatal harm on panc-1 in radiofrequency field	Panc-1	[127]
C225-AuNP	13.56	600	32.6 ± 0.7	Heterologous PC grafts were notably disrupted without evident treatment toxicity	Panc-1/Capan-1	[126]

**Table 3 pharmaceutics-16-00806-t003:** Studies and results of AuNPs in radiotherapy for PC.

Nanoparticles	Outcome	Cell Lines	Ref
Au-MIP microgels	Tumor growth in mice was effectively inhibited	MIAPaCa-2	[131]
AuNP	The size of tumors and cellular proliferation significantly decreased	MIAPaCa-2	[132]
LNPDTX-P	The intake of AuNPs significantly increased	MIAPaCa-2	[133]
gold ultra-small nanoparticles	Enhanced DNA damage and cell apoptosis led to delayed tumor growth	MIAPaCa-2/SUIT2-028	[134]
211At-AuNPs@mPEG	The prolonged retention of 211At in PC tissues results in notable antitumor activity	Panc-1	[137]

## Data Availability

Not applicable.

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
