# Peer review of "Prospect of Gold Nanoparticles in Pancreatic Cancer"

_pharmaceutics, 2024, doi:10.3390/pharmaceutics16060806_

Round 1
Reviewer 1 Report
Comments and Suggestions for Authors
In the review manuscript “Prospect of gold nanoparticles in pancreatic cancer” Yin et al. reviewed the application and future directions of AuNPs in the diagnosis and treatment of PC. The manuscript is well-written and structured, the figures are effectively aid in illustrating key concepts and the review demonstrates a thorough understanding of the subject. Nonetheless, there are several areas where further expansion would significantly enhance the depth and utility of the review, as detailed in the file.

The manuscript is well-written.
Reviewer 2 Report
Comments and Suggestions for Authors
The article “Prospect of gold nanoparticles in pancreatic cancer” by Tianyi Yin, et al, is a peer-review article on using Gold nanoparticles for the potential therapy of PC. The article is well-written, precise and coherent, addressing all the relevant issues, including toxicity. The last is mainly a severe drawback to applying NPs for cancer therapies because localized extracellular interactions between nanoparticles and transmembrane signal receptors may well activate cancer cell growth via activation of AKT and ERK signalling pathways and mechanical stimulation of ligand adhesion binding sites of integrins and EGFR. The authors should describe in detail the NPs toxity in the text (DOI: 10.1186/s11671-018-2775-z).The article can be published after the revision.
Reviewer 3 Report
Comments and Suggestions for Authors
This review comprehensively explores the application of gold nanoparticles (AuNPs) in both the diagnosis and treatment of pancreatic cancer. The article highlights the enhanced efficacy of various treatment methods facilitated by the incorporation of AuNPs.
To improve the manuscript, I recommend incorporating additional schemes and figures for sections 3 and 4, as well as for subsections 5.2, 5.3, and 5.4.
Furthermore, the conclusions could be extended to include insights into future perspectives by discussing the trajectory of this research field, potential milestones, and anticipated challenges.
Reviewer 4 Report
Comments and Suggestions for Authors
In this manuscript, Yin and colleagues described the application and future directions of gold nanoparticles in the diagnosis and treatment of pancreatic cancer, along with their clinical challenges. This manuscript also covers the characteristics, preparation techniques, diagnostic applications in pancreatic cancer, and the safety aspects of the gold nanoparticles.
The manuscript was interesting to read. Some changes will have to be made, as described below:
1. The abstract is a bit vague; it would be good to further explain the rationale behind the use of gold nanoparticles for the diagnosis and treatment of this disease.
2. The various parts of the review should be further discussed.
3. Line 10: “Pancreatic cancer (PC) is also called "king of cancer"”: could you please re-write this in a more scientific way?
4. Line 23: “Owing to the absence of 23 effective screening methods for PC”: could you please also describe the symptoms of the disease, that could explain why the diagnosis is done at an advanced stage?
5. Line 27: “PC research is in a bottleneck period”: could you please re-write this in a more scientific way?
6. Line 30: could you please give the full name of EUS?
7. Line 45: “AuNPs have rapidly advanced”: could you please reformulate this?
8. Figure 1: Could you please re-draw the graphical abstract in a more scientific way?
9. Line 97: could you please format the Latin name of the plants in italic?
10. Line 99: Could you please correct the formatting of HAuCl4 and NaAuCl4 in the whole manuscript?
11. Line 105: “Proper staging of PC at diagnosis”: could you please reformulate this?
12. Line 123: could you please format in vitro, in vivo in italic in the whole manuscript?
13. Line 179: “Nonetheless, the effectiveness of GEM in treating PC is less than desirable”: could you please reformulate this?
14. Line 181: “numerous nanodelivery systems like liposomes”: could you please replace “like” by “such as “?
15. Line 188: “A drug delivery system comprising”: could you please reformulate this?
16. Line 336: The safety part should be further discussed.
17. Line 359: “Nonetheless, the potential toxicity and safety concerns associated with AuNPs warrant further investigation”: this is contradictory, could you please reformulate this?
18. Line 374: could you please format the publications using the same format (i.e. name of the journals in full or abbreviated for all the references)
Comments on the Quality of English Language
The style of the manuscript should be carefully checked and modified when needed. Some specific examples are provided in my comments above.
